

# Genome-wide QTL mapping of yield and agronomic traits in two widely adapted winter wheat cultivars from multiple mega-environments

Smit Dhakal[1], Xiaoxiao Liu[1], Chenggen Chu[1,2], Yan Yang[1], Jackie C. Rudd[1], Amir M.H. Ibrahim[3], Qingwu Xue[1], Ravindra N. Devkota[1], Jason A. Baker[1], Shannon A. Baker[1], Bryan E. Simoneaux[3], Geraldine B. Opena[3], Russell Sutton[3], Kirk E. Jessup[1], Kele Hui[1], Shichen Wang[4], Charles D. Johnson[4], Richard P. Metz[4] and Shuyu Liu[1]

[1] Texas A&M AgriLife Research and Extension Center, Texas A&M AgriLife Research, Amarillo, TX, United States of America
[2] Edward T. Schafer Agricultural Research Center, Sugarbeet & Potato Research Unit, USDA-ARS, Fargo, ND, United States of America
[3] Department of Soil and Crop Sciences, Texas A&M University, College Station, TX, United States of America
[4] Genomics and Bioinformatics Service Center, Texas A&M AgriLife Research, College Station, TX, United States of America

Corresponding author
Shuyu Liu, SLiu@ag.tamu.edu

## ABSTRACT

Quantitative trait loci (QTL) analysis could help to identify suitable molecular markers for marker-assisted breeding (MAB). A mapping population of 124 $F_{5:7}$ recombinant inbred lines derived from the cross 'TAM 112'/'TAM 111' was grown under 28 diverse environments and evaluated for grain yield, test weight, heading date, and plant height. The objective of this study was to detect QTL conferring grain yield and agronomic traits from multiple mega-environments. Through a linkage map with 5,948 single nucleotide polymorphisms (SNPs), 51 QTL were consistently identified in two or more environments or analyses. Ten QTL linked to two or more traits were also identified on chromosomes 1A, 1D, 4B, 4D, 6A, 7B, and 7D. Those QTL explained up to 13.3% of additive phenotypic variations with the additive logarithm of odds (LOD(A)) scores up to 11.2. The additive effect increased yield up to 8.16 and 6.57 g m$^{-2}$ and increased test weight by 2.14 and 3.47 kg m$^{-3}$ with favorable alleles from TAM 111 and TAM 112, respectively. Seven major QTL for yield and six for TW with one in common were of our interest on MAB as they explained 5% or more phenotypic variations through additive effects. This study confirmed previously identified loci and identified new QTL and the favorable alleles for improving grain yield and agronomic traits.

## INTRODUCTION

Grain yield in wheat (*Triticum aestivum* L.) is a major goal of most of the wheat breeding programs, particularly in rainfed growing areas (*Brinton et al., 2017*). In the Southern High Plains, bread wheat is grown under a wide range of mega environments that differ for soil

moisture and rainfall pattern. Besides genetic and environmental factors, grain yield is also influenced by agronomic and morphological traits like heading date and plant height (*Chen et al., 2012*; *Liu et al., 2015*). Genetic gains in grain yield have been attributed to the development and deployment of high-yielding wheat varieties with improved agronomic traits related to high yield potential (*De Vita et al., 2007*; *Gao et al., 2017*; *Lopes et al., 2012*). Due to their high heritabilities and correlations with grain yield, agronomic traits such as heading date and plant height are important traits to be considered during breeding and cultivar development (*Chen et al., 2012*; *Gao et al., 2017*; *Liu et al., 2015*).

Plant height is controlled by many reduced height (*Rht*) genes that play roles on reducing the length of coleoptile and internode and thus decrease plant height (*Rebetzke et al., 2012*). Higher grain yield can be achieved by reducing the internal competition to increase assimilate partitioning to the economic sinks and the straw strength (*Addisu et al., 2010*; *Borlaug, 1968*; *Grover et al., 2018*; (*Worland, 1996*)). Heading date is influenced by *Vrn*, *Ppd* and *Efl* genes governing vernalization, photoperiod response, intrinsic earliness, and their interactions (*Mondal et al., 2016*; *Scarth & Law, 1984*; *Sourdille et al., 2000*; *Worland et al., 1998*). Test weight (TW, also called grain volume weight) is a volumetric measurement (mass/volume) determined by weighing grain samples filled in a standard dry one-quart measure. TW is a trait directly associated with grain quality and an indirect indicator of seed size and shape that ultimately affects kernel weight (*Campbell et al., 1999*; *Juliana et al., 2019*).

Grain yield and agronomic traits were significantly affected by genotype, environment, and genotype-by-environment interactions, which mainly influenced the genetic gain achieved by phenotypic selection. Change in the related performance of lines across environments further complicates selection. Understanding the effects of genetic and genotype-by-environment interaction on yield-related traits can enhance yield improvement during cultivar development (*Dhungana et al., 2007*; *Xing & Zhang, 2010*). With the utilization of multi-environment trials and a high-density genetic map covering all chromosomes, QTL mapping has enabled the dissection of complicated traits like grain yield into individual loci, as well as the ability to quantify epistasis effects among different loci and QTL-by-environment interactions (*Doerge, 2002*). Validated diagnostic markers associated with targeted QTL can be used in marker-assisted selection. Particularly, breeder-friendly markers linked to QTL associated with agronomic traits will allow breeders to understand the genetic architecture of germplasms, target interested gene loci, and assign heterotic pools in hybrid wheat breeding programs (*Adhikari et al., 2020a*; *Adhikari et al., 2020b*).

In the current research, grain yield and important agronomic traits of 124 recombinant inbred lines (RILs) derived from two most widely grown hard red winter wheat cultivars in the Southern Great Plains were characterized in multi-environment trials. Linkage and QTL analyses were conducted to identify genomic regions controlling grain yield and related agronomic traits. The complex genetic basis of four traits was dissected and interpreted.

## MATERIALS & METHODS

### Plant materials, field trials and phenotyping

A population of 124 $F_{5:7}$ RILs derived from the cross 'TAM 112'/'TAM 111' was used to map grain yield and related agronomic traits. Both TAM 111 and TAM 112 were developed by Texas A&M AgriLife Research (*Lazar et al., 2004*; *Rudd et al., 2014*) and well-adapted hard red winter wheat (HRWW) cultivars in the Southern Great Plains of the United States. TAM 112 showed better adaption to low input environments, whereas TAM 111 typically produced higher yield in moderate to high input environments. The RILs along with the two parents were planted in five locations including Bushland, TX (35°06′N, 102°27′W), Chillicothe, TX (34°15′N, 99°30′W), Clovis, NM (34°24′N, 103°12′W), Etter, TX (35°51′N, 101°58′W), and Uvalde, TX (29°21′N, 99°75′W) during five years (2011 to 2014, and 2017) (Table 1). Yield data were collected from 28 year-location-management combinations (environments) which included Bushland dryland (BD) in 2011 (11BD), 2012 (12BD) and 2017 (17BD), Bushland irrigated (BI) in 2017 (17BI), Chillicothe dryland (CH) in 2011 (11CH), 2012 (12CH), and 2014 (14CH), Clovis irrigated (CVI) in 2017 (17CVI), Etter with linear irrigation in 2017 (17EI), and various irrigation levels in Etter and Uvalde. Etter with five irrigation levels as 40% of evapotranspiration (ET) demand (EP1), 50% of ET demand (EP2), 65% of ET demand (EP3), 75% of ET demand (EP4), 100% of ET demand (EP5) during 2011–2014 included 14 environments (11EP1, 11EP2, 11EP3, 11EP4, 11EP5, 12EP1, 12EP2, 12EP3, 13EP2, 13EP3, 13EP4, 13EP5, 14EP4, and 14EP5). Uvalde included dryland (UVD) and Uvalde at three irrigation levels at 50% (UV5), 70% (UV7), and 100% of ET demand (UVL) in 2012 and 2013 (12UVD, 12UV5, 12UV7, 12UVL, and 13UVL). Overall, sixteen environments (11BD, 12BD, 17BD, 11CH, 12CH, 14CH, 11EP1, 11EP2, 11EP3, 12EP1, 12EP2, 12EP3, 13EP2, 13EP3, 12UVD, and 12UV5) were considered as dryland condition and twelve others (17BI, 17CVI, 11EP4, 11EP5, 13EP4, 13EP5, 14EP4, 14EP5, 17EI, 12UV7, 12UVL, and 13UVL) were considered as irrigated conditions. All trials were replicated twice in an alpha-lattice design with an incomplete block size of five plots and each parent occurring three times in each replication. The plot dimension was 6.09 m × 1.52 m in the dryland environments and 4.57 m × 1.52 m in the irrigated ones with a 0.3-m space between plots. Standard agronomic practices were performed in each trial (*Dhakal et al., 2021*; *Yang et al., 2020b*).

All 28 environments were harvested using a combine harvester and the total plot weight was used to calculate grain yield (YLD). Traits of test weight (TW), heading date (HD), and plant height (HT) were measured in a subset of environments. TW was measured using Seedburo equipment (http://www.seedburo.com, Des Plaines, IL, USA) from 19 environments (11BD, 17BD, 17BI, 12CH, 14CH, 11EP5, 12EP1, 12EP2, 12EP3, 13EP2, 13EP3, 13EP4, 13EP5, 14EP4, 14EP5, 17EP5, 12UV5, 12UV7, and 12UVL). HD was recorded at Feekes growth stage 10.1 when half of the plants were fully visible on heads from 11 environments (11BD, 12BD, 17BD, 11EP1, 11EP2, 11EP3, 11EP4, 11EP5, 12EP1, 12EP2, and 12EP3). Plant height was measured in centimeters (cm) from representative plants in each plot as the distance from the base of the stem to the top of the spike excluding

Dhakal et al. (2021), *PeerJ*, DOI 10.7717/peerj.12350

**Table 1** Locations, cropping seasons, geographic coordinates, and climatic characterization of trials with growing seasons ending in year 2011, 2012, 2013, 2014 and 2017 at five different locations.

| Location | Altitude (masl) | Latitude | Longitude | Season | Irrigation[a] | Environments[b] | Seasonal Temperature (°C)[c] | | | Rainfall[d] (mm) |
|---|---|---|---|---|---|---|---|---|---|---|
| | | | | | | | Max$_{av}$ | Av. | Min$_{av}$ | |
| Bushland, TX | 1,098 | 35°16′N | 102°27′W | 2010/2011 | $I_0$ | 11BD | 16.83 | 11.78 | 3.11 | 128.27 |
| | | | | 2011/2012 | $I_0$ | 12BD | 20.33 | 12.56 | 4.78 | 265.94 |
| | | | | 2016/2017 | $I_0$, $I_{100}$ | 17BD, 17BI | 21.17 | 12.83 | 4.56 | 275.59 |
| Chillicothe, TX | 436 | 34°15′N | 99°30′W | 2010/2011 | $I_0$ | 11CH | 24.33 | 15.83 | 6.33 | 59.18 |
| | | | | 2011/2012 | $I_0$ | 12CH | 24.44 | 16.11 | 9.33 | 230.12 |
| | | | | 2013/2014 | $I_0$ | 14CH | 23.61 | 13.67 | 2.5 | 197.61 |
| Clovis, NM | 1,309 | 34°24′N | 103°12′W | 2016/2017 | $I_{100}$ | 17CVI | 20.17 | 12.89 | 3.06 | 197.87 |
| Etter, TX | 1,117 | 35°51′N | 101°58′W | 2010/2011 | $I_{40}$, $I_{50}$, $I_{65}$, $I_{75}$, $I_{100}$ | 11EP1, 11EP2, 11EP3, 11EP4, 11EP5 | 19.6 | 11.04 | 1.3 | 62 |
| | | | | 2011/2012 | $I_{40}$, $I_{50}$, $I_{65}$ | 12EP1, 12EP2, 12EP3 | 19.4 | 11.48 | 2.9 | 243 |
| | | | | 2012/2013 | $I_{50}$, $I_{65}$, $I_{75}$, $I_{100}$ | 13EP2, 13EP3, 13EP4, 13EP5 | 18.8 | 10 | 1.4 | 81 |
| | | | | 2013/2014 | $I_{75}$, $I_{100}$ | 14EP4, 14EP5 | 17.8 | 9.19 | 0.6 | 138 |
| | | | | 2016/2017 | $I_{100}$ | 17EI | 19.6 | 9.9 | −0.7 | 152.6 |
| Uvalde, TX | 378 | 29°21′N | 99°75′W | 2011/2012 | $I_0$, $I_{50}$, $I_{75}$, $I_{100}$ | 12UVD, 12UV5, 12UV7, 12UVL | 25.83 | 19.17 | 11.39 | 253.75 |
| | | | | 2012/2013 | $I_0$, $I_{100}$ | 13UVD, 13UVL | 25.78 | 19.11 | 11 | 262.13 |

**Notes.**

[a] Irrigation levels: I0, no irrigation; I40, 40% field capacity of irrigation; I50, 50% field capacity of irrigation; I65, 65% of field capacity irrigation, I75, 75% of field capacity irrigation; $I_{100}$, 100% of field capacity irrigation.

[b] Abbreviations: 11, Year 2011; 12, Year 2012; 13, Year 2013; 14, Year 2014; 17, Year 2017; BD, Bushland dry ($I_0$), TX; BI, Bushland Irrigated ($I_{100}$), TX; CH, Chillicothe ($I_0$), TX; CVI, Clovis Irrigated ($I_{100}$), NM; EP1, Etter ($I_0$), TX; EP2, Etter ($I_{50}$), TX; EP3, Etter ($I_{65}$), TX; EP4, Etter ($I_{75}$), TX; EP5, Etter (100), TX; UVD, Uvalde dry ($I_0$), TX; UV5, Uvalde ($I_{50}$), TX; UV7, Uvalde ($I_{70}$), TX; UVL, Uvalde ($I_{100}$), TX.

[c] Seasonal Temperatures includes readings from plant in October of the previous years to harvesting in June of the current year: Maxav, average of the maximum temperatures during growing seasons; Av, average of the average temperature during growing seasons; Minav, average of the minimum temperature during growing seasons.

[d] Cumulative rainfall during the growing seasons.

awns at maturity in 11 environments (11BD, 12BD, 17BD, 12CH, 17CVI, 11EP1, 11EP2, 11EP3, 11EP4, 11EP5, and 17EI).

## Statistical analysis

Descriptive statistics were calculated using PROC UNIVARIATE in SAS 9.4 (SAS Institute, Cary, NC, USA). The histograms of the residuals for all traits were approximately normal. Best linear unbiased prediction (BLUP) means of lines were calculated for the agronomic traits from the individual environment (IE) as well as multi-environment trials (MET). BLUP was calculated using a restricted maximum likelihood (REML) approach implemented in the 'lme4' package through META-R (*Alvarado et al., 2018*).The analysis of variance (ANOVA) for IE followed a linear statistical model of individual environment analysis with replication and incomplete block as random effects, while for MET linear statistical model of combined environment analysis was run using PROC MIXED in SAS 9.4 with the environment, replication and incomplete block as random effects. The entry-mean heritabilities, pearson's correlations, biplot analyses, mega-environments classifications followed the same procedures (*Dhakal et al., 2021*).

## Genotyping and linkage map construction

DNA extraction, Illumina Infinium iSelect 90K array SNP genotyping and Genome Studio clustering followed the procedures described in the literature (*Assanga et al., 2017*; *Dhakal et al., 2018*; *Liu et al., 2016*; *Yang et al., 2019*). This population was also genotyped with double digest restriction-site associated DNA sequencing (ddRADSeq) method developed by *Peterson et al. (2012)* on an Illumina HiSeq 2500 platform ($2 \times 125$ bp paired-end) following the standard procedures (*Yang et al., 2020b*). JoinMap v4.0 software (*Van Ooijen, 2006*) was used to construct the genetic maps using the standard procedures (*Dhakal et al., 2021*; *Yang et al., 2020b*). The SNP sequences were listed in Table S1.

## QTL analysis

The additive effects, epistasis, additive-by-environment, and epistasis-by-environment interactions were analyzed using the integrated composite interval mapping (ICIM) function implemented in QTL IciMapping software (*Meng et al., 2015*). The threshold for declaring a QTL was determined through a permutation test ($n = 1,000$) for a single environment to obtain a 0.05 genome-wide probability level of Type I error. QTL analyses were run for individual environment (IE) for additive effects (ADD), multiple environment (EPI), and within and across MEs following the standard procedures (*Dhakal et al., 2021*). QTL designation followed *McIntosh et al. (2003)* guidelines with a slight change using the format *Qtrait.tamu.chrom.Mb*, where *trait* represents a trait name, *tamu* indicates Texas A&M University, *chrom* means the chromosome harboring the QTL, and *Mb* indicates the Mega base pair (Mb) position of the peak SNP within a QTL according to sequence alignment using the IWGSC RefSeq v1.0 reference genome (*International Wheat Genome Sequencing Consortium, 2014*).

## RESULTS

### Phenotypic variability, heritability, and correlations

Each environment was unique and different from the others across years. In general, Chillicothe and Uvalde, TX were warmer and Etter, TX located in the northern Texas High Plains tends to be cooler. Bushland received below average rainfall in 2011 and almost double in 2012 and 2017; however, the rainfalls during the wheat growing seasons were low during 2011-2014, which were all considered drought years for wheat (Table 1). Temperature was within the optimum range required for winter wheat growth. Combined ANOVA showed highly significant genotype differences ($p < 0.001$) among RILs for all the traits ($p < 0.01$) (Table 2). All the traits were highly heritable (0.77–0.96). Means for YLD, TW, HD, and HT were 287.7 g m$^{-2}$, 760.6 kg m$^{-3}$, 115.5 Julian days, and 73.8 cm, respectively. Transgressive segregation was observed for all the traits. Based on the means across all environments, TAM 112 had higher yield while TAM 111 had higher TW, HD and HT based on the overall BLUP means from all tested environments. However, the two parents did not differ significantly for all traits (Table 2).

The phenotypic relationships between grain yield and agronomic traits in individual environment was determined using Pearson's correlation coefficients (Table S2). In general, negative correlations were found between HD and YLD in the dryland environments with high correlation values at some dryland environments. Most correlations between YLD and HT were positive and significant in the dryland environments, especially in 2011. Most correlations between YLD and TW were positive and significant except for a few dryland environments. HD and HT were significantly and negatively correlated in dryland environments except for 17BD as 2017 was a high rainfall year while they were positively correlated in highly irrigated environments. TW had negative correlations with HD while it had positive correlations with HT in dryland environments but negatively correlated in irrigated environments with limited amount of data (Table S2).

### Boxplot, biplot, and mega-environment

Heading date was later for the year 2011 except the 11BD environment. Earlier HD was seen in 11BD, 12BD, and 17BD (Fig. S1). Plant height was severely affected by drought, as shown in the Bushland dryland and less affected under irrigated environments in Etter. Plant height was less affected by the year with high rainfall such as 2017. Plants were taller in irrigated and higher rainfall environments (17CVI and 17EI), almost three times taller in these optimum environments. Test weight was also affected by drought. The lowest TW was observed in 12EP1 and 12EP3, while the highest TW was in 17BI. Grain yield was lowest in the driest year and dry environments. However, the top 20 high yielding lines across individual environments were distributed within the 106 lines out of 124 lines (Table S3), showing the strong genotype-by-environment interactions and the necessary for mega-enviroement classification.

Mega-environment analysis categorized all the environments into two or four different mega-environments for different traits (Fig. 1). Grain yield had ME1 (11CH, 14EP4, 14EP5, 17BI, and 17CVI), ME2 (11BD, 11EP5, 12BD, 12EP1, 12EP2, and 17EI), ME3 (12CH, 12UVD, 13UVL, and 17BD), and ME4 (12UV5, 12UV7, 12UVL, and 14CH);

Dhakal et al. (2021), *PeerJ*, DOI 10.7717/peerj.12350

**Table 2** **Analysis of variance, heritability and mean performance of grain yield and agronomic traits.**

| Traits | Units | $\sigma^2_{Geno}$[a] | $\sigma^2_{Env}$[b] | $\sigma^2_{Rep(Env)}$[c] | $\sigma^2_{Iblk(Rep*Env)}$[d] | $\sigma^2_{GE}$[e] | $\sigma^2_{Res}$[f] | RILs | | | Parents | |
|---|---|---|---|---|---|---|---|---|---|---|---|---|
| Traits | | | | | | | | $h^2_g$ | LSD | X ± SD | TAM 112 | TAM 111 |
| Grain yield (YLD) | g m$^{-2}$ | 265.6** | 17972.0** | 281.5** | 575.6** | 853.3** | 795.7** | 0.77 | 55.3 | 287.7 ± 139.4 | 304.4 | 298.6 |
| Test weight (TW) | kg m$^{-3}$ | 144.1** | 538.3** | 13.2 | 21.5 | 135.8** | 26.5** | 0.94 | 10 | 760.6 ± 27.0 | 767 | 770.1 |
| Heading Date (HD) | Days from Jan 1 | 3.05** | 15.46** | 0.01 | 0.21** | 1.30** | 1.28** | 0.96 | 2.22 | 115.5 ± 4.01 | 113.68 | 117.36 |
| Plant Height (HT) | cm | 4.31** | 404.71** | 0.83 | 2.41** | 3.48** | 13.16** | 0.87 | 7.11 | 73.8 ± 19.01 | 72.74 | 73.95 |

**Notes.**

[a] $\sigma^2_{Geno}$, genotypic variance.

[b] $\sigma^2_{Env}$, variance due to environment.

[c] $\sigma^2_{Rep}$ (Env), variance due to replication nested within environments.

[d] $\sigma^2_{Iblk}$ (Rep*Env), variance due to incomplete block nested within replication (environment).

[e] $\sigma^2_{GE}$, variance due to genotype-by-environment interaction.

[f] $\sigma^2_{Res}$, residual variance.

[g] $h^2$, entry-mean heritability.

*, ** significant at 0.05, and 0.01 probability levels, respectively.

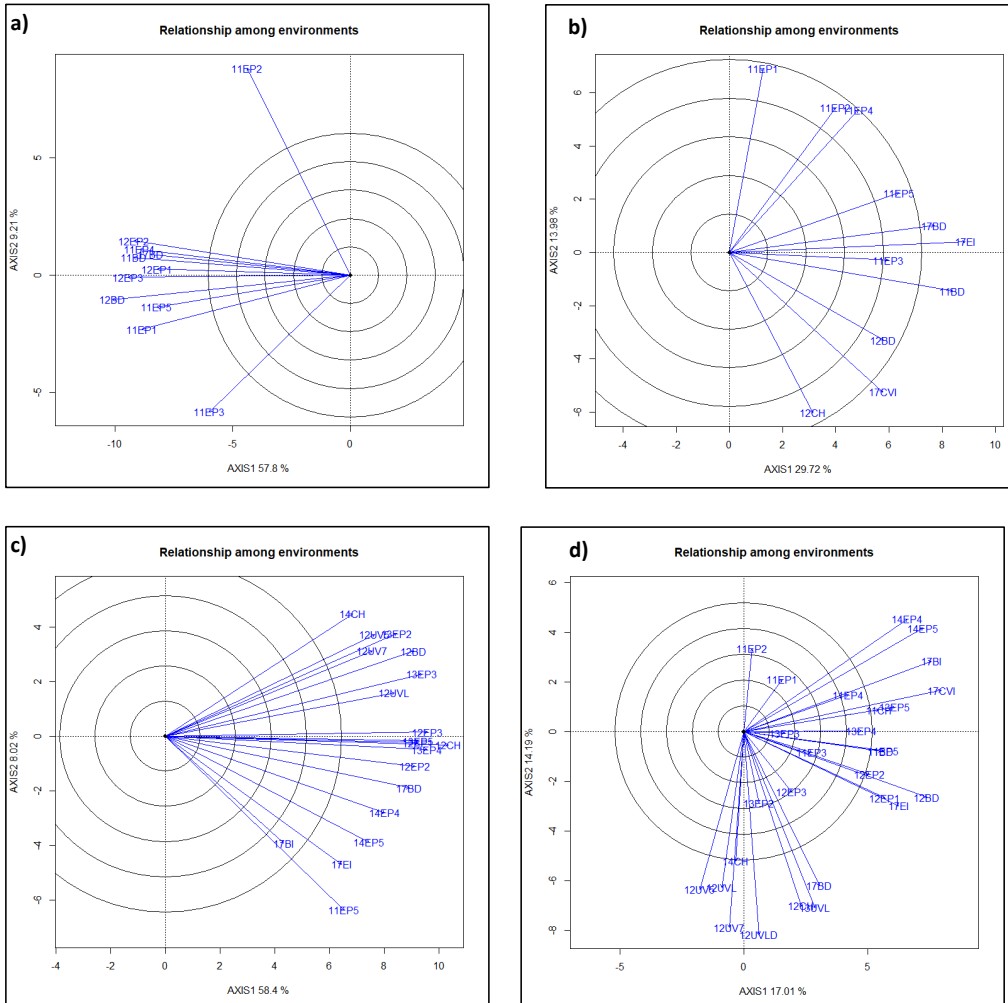

**Figure 1** **GGE-Biplot analysis of yield and agronomic traits to classify mega-environments for each trait. Clusters of environments within a quadrant is in a mega-environment.** ME1, ME2, ME3, ME4 are mega-environment 1, 2, 3, and 4 where lines performed very similarly. Individual Environments were Bushland, TX in 2011, 2012, 2017 as dryland (11BD, 12BD, 17BD), irrigated in 2017 (17BI), Chillicothe, TX in 2011, 2012 and 2014 as dryland (11CH, 12CH, 14CH), Etter, TX with five irrigation levels (40%, 50%, 65%,75% and 100%) in 2011 (11EP1, 11EP2, 11EP3, 11EP4, 11EP5), three irrigation levels in 2012 (12EP1, 12EP2, 12EP3), four irrigation levels in 2013 (13EP2, 13EP3, 13EP4, 13EP5), two irrigation levels in 2014 (14EP4, 14EP5), and irrigated in 2017 (17EI), Uvalde, TX in 2012 as dryland (12UVLD = 12UVD), and with three irrigation levels in 2012 (50%, 75%, 100%) (12UV5, 12UV7, 12UVL), irrigated in 2013 (13UVL), Clovis, NM irrigated in 2017 (17CVI). ME's for traits were (A)) heading days, (B) plant height, (C)) test weight, ME1 included 12BD, 12EP2, 12UV7, 12UVL, 12UV5, 13EP3, and 14CH; ME2 included 11EP5, 12CH, 12EP1, 12EP3, 13EP2, 13EP4, 13EP5, 14EP4, 14EP5, 17BD, 17BI, and 17EI; (D) grain yield, ME1 included 11CH, 14EP4, 14EP5, 17BI, and 17CVI; ME2 included 11BD, 11EP5, 12BD,12EP1, 12EP2, 17EI; ME3 included 12CH, 12UVD, 13UVL,17BD; ME4 included 12UV5, 12UV7, 12UVL, and 14CH.

Test weight had ME1 (12BD, 12EP2, 12UV7, 12UVL, 12UV5, 13EP3, and 14CH) and ME2 (11EP5, 12CH, 12EP1, 12EP3, 13EP2, 13EP4, 13EP5, 14EP4, 14EP5, 17BD, 17BI, and 17EI). Plant height had ME1 (11EP3, 11BD, 12BD, 17CVI, and 12CH) and ME2 (11EP1, 11EP2, 11EP4, 11EP5, 17BD, and 17EI). Heading date had ME1 (12EP2, 11EP4, 11BD, 12EP1, and 17BD) and ME2 (12EP3, 12BD, 11EP5, 11EP1, and 11EP3). 11EP2 was excluded since it was far deviated from the rest (Fig. 1). QTL was analyzed within and across ME's for yield and test weight.

## Linkage map and QTL analyses

We used JoinMap 4.0 and QTL ICIMapping 4.2 to construct genetic maps and QTL analyses following similar procedures as *Yang et al. (2019)*.

A set of 115 unique QTL regions significantly associated with YLD and related agronomic traits across 28 environments over five years was identified through the analyses of data from IE, MET, ME (Table 3, Table S4). Among them, 51 unique consistent QTL were associated with a single trait from at least two out of IE, MET, ME analyses (Table 3, Fig. 2, Figs. S2, and S3). These consistent QTL were identified for all the four traits analyzed on 15 chromosomes including 1A, 1D, 2B, 2D, 3B, 3D, 4B, 4D, 5A, 5B, 6A, 6B, 6D, 7B, and 7D.

## Grain yield

A total of 18 QTL for YLD were detected. Nine common consistent and pleiotropic QTL were located on chromosomes 1D, 4B, 4D, 6A, 7B, and 7D. The other nine consistent QTL were mapped on chromosomes 1A, 3B, 4B, 4D, and 7D (Table 3, Table S4, and Fig. 2). Favorable alleles from TAM 111 for QTL on the chromosomes 1A (411.7 and 585.6 Mb), 3B (48.6 Mb), 4B (226.8 Mb), 4D (445.5 Mb), 6A (12.4 and 608.5 Mb), 7B (15.6 Mb), and 7D (84.3 Mb) explained up to 41.3% of total phenotypic variations and increased yield by up to 37.41 g m$^{-2}$ from *Qyld.tamu.4B.267* in 17CVI while the remaining QTL with favorable alleles from TAM 112 increased yield by up to 13.54 g m$^{-2}$ and explained a total of 25% of yield variations at ME3 including 12CH, 12UVD, 13UVL, and 17BD from *Qyld.tamu.1A.587* (Table 3, Table S4, and Fig. S3). Favorable allele switched between two parents for QTL on the chromosome 4B at 266.8 Mb and 4D at 445.5 Mb. TAM 111 favorable allele of *Qyld.tamu.4B.267* increased yield by 37.41 g m$^{-2}$ in 17CVI and increased yield by 3.38 g m$^{-2}$ across ME1, including 11CH, 14EP4, 14EP5, 17BI, and 17CVI while TAM 112 allele only increased yield by 0.42 g m$^{-2}$ when analyzed across all 28 environments. However, additive-by-environment contributed very large part of PVE. The additive-by-17CVI interaction explained 63.3% out of the total 63.5% PVE (Table S4). The QTL *Qyld.tamu.4D.446* had favorable alleles from TAM 111 that increased yield by 8.59 and 3.98 g m$^{-2}$ when analyzed for individual environment in 14CH and across ME4, including 12UV5, 12UV7, 12UVL, and 14CH while it had favorable allele from TAM 112 that increased yield by 0.47 g m$^{-2}$ when analyzed across all 28 environments (Table 3, Table S4).

Among the major consistent QTL for yield, those QTL that had larger additive effect contributions from across environment analyses are of interest to breeders for yield

**Table 3  Consistent and pleiotropic QTL for grain yield and agronomic traits detected from individual and multiple environment QTL analysis.**

| QTL (underlined involved interactions) | Chr | Position[a] (Mb) | Trait[b] | Environment[c] | Linkage[d] | Peak[e] (cM) | QTL CI[f] (cM) | LOD[g] | LOD (A) | LOD (A*E) | PVE[h] (%) | PVE (A) (%) | PVE (A*E) (%) | ADD[i] | SNP alleles increase traits | Left SNPs alleles[j] | Right SNPs alleles[k] | Pleiotropic QTL | Citation for known QTL |
|---|---|---|---|---|---|---|---|---|---|---|---|---|---|---|---|---|---|---|---|
| *Qyld.tamu.1A.412* | 1A | 411.7 | YLD | 17EI, AcrossME2 | 1A | 58 | 57.5–58.5 | 5.2–6.8 | 0.4 | 6.4 | 8.3–12.9 | 0.8 | 7.5 | −1.42-(−10.22) | TAM 111 | C/T | G/A | | *Yang et al. (2020b)* |
| *Qyld.tamu.1A.586* | 1A | 585.6 | YLD | ME3, Across ME1234 | 1A | 172 | 171.5–172.5 | 24.9–25.2 | 2.8 | 22.4 | 16.8–21.2 | 2.8 | 18.4 | −1.94-(−11.1) | TAM 111 | C/T | A/G | | *Adhikari et al. (2020a)*; *Juliana et al. (2019)* |
| *Qyld.tamu.1A.587* | 1A | 587.0 | YLD | ME3, Across ME1234 | 1A | 174 | 173.5–175 | 34.9–36.3 | 12.0 | 24.3 | 12.0–25.0 | 14.6 | 20.6 | 4.43–13.54 | TAM 112 | A/T | T/G | | *Alvarez et al. (2016)* and *Zhang et al. (2018)* |
| *Qyld.tamu.1D.422* | D3B | 421.8 | YLD | 12CH, across all env, AcrossME3 | 1D | 69 | 68.5–69.5 | 5.6–17.0 | 3.1–3.7 | 3.2–13.9 | 5.6–12.7 | 3.6–4.2 | 1.4–9.1 | 1.86–6.87 | TAM 112 | T/C | C/A | TW | *Yang et al. (2020b)* |
| *Qyld.tamu.3B.49* | 3B | 48.6 | YLD | 12UVL, AcrossME4 | 3B | 5 | 4.5–5.5 | 5.0–7.5 | 3.2 | 4.4 | 9.5–9.8 | 4.4 | 5.0 | −6.75-(−17.58) | TAM 111 | G/A | G/C | | |
| *Qyld.tamu.4B.267* | 4B | 266.8 | YLD | 17CVI, across all env, AcrossME1 | 4B | 39 | 38.5–39.5 | 20.4–28.1 | 0.2–1.5 | 22.2–27.9 | 22.3–63.5 | 0.2–1.3 | 35.0–63.3 | 0.42-(−37.41) | TAM 112, TAM 111 | G/A | C/T | | |
| Qyld.tamu.4B.659 | 4B | 659.2 | YLD | 14EP4, AcrossME1 | 4B | 94 | 93.5–94.5 | 3.6–6.8 | 5.6 | 1.2 | 6.5–9.8 | 4.9 | 1.6 | 6.43–9.88 | TAM 112 | A/G | A/G | HD | *Yang et al. (2020b)* |
| *Qyld.tamu.4B.661* | 4B | 660.9 | YLD | 12UVLD, AcrossME3 | 4B | 99 | 97.5–100.5 | 4.3–5.3 | 3.0 | 2.3 | 8.4–11.9 | 3.3 | 5.0 | 3.28–9.65 | TAM 112 | T/C | G/A | | *Yang et al. (2020a)* |
| Qyld.tamu.4D.21 | 4D | 20.6 | YLD | 17CVI, AcrossME1, ME1, Across ME1234 | 4D | 1 | 0–4.5 | 5.4–8.4 | 4.1–6.4 | 2.1–2.2 | 4.9–19.4 | 4.8–5.5 | 3.1–8.5 | −2.55-(−17.32) | TAM 111 | A/G | T/C | HT | *Yang et al. (2020b)* |
| *Qyld.tamu.4D.110* | 4D | 109.8 | YLD | 17BI, across all env, AcrossME1 | 4D | 13 | 12.5–13.5 | 8.3–18.0 | 2.4–8.6 | 3.1–15.6 | 18.4–41.3 | 2.8–7.9 | 12.4–38.5 | −1.64-(−28.06) | TAM 111 | C/T | T/C | TW | *Dhakal et al. (2021)* and *Yang et al. (2020b)* |
| *Qyld.tamu.4D.446* | 4D | 445.5 | YLD | 14CH, across all env, AcrossME4 | 4D | 28 | 27.5–28.5 | 13.4–21.2 | 0.2–1.1 | 12.7–21.0 | 2.3–28.2 | 0.2–1.6 | 0.8–12.7 | −0.47-8.59 | TAM 111 | T/C | A/G | | |
| Qyld.tamu.6A.12 | 6A | 12.4 | YLD | 17BI | 6A | 20 | 19.5–20.5 | 4.4 | | | 8.9 | | | −19.58 | TAM 111 | G/C | T/G | TW | *Yang et al. (2020b)* |
| *Qyld.tamu.6A.609* | 6A | 608.5 | YLD | ME3, Across ME1234 | 6A | 139 | 138.5–139.5 | 3.7–5.4 | 0.1 | 5.3 | 1.4–5.2 | 0.1 | 5.0 | −0.42-(−3.18) | TAM 111 | T/C | T/C | TW | |
| *Qyld.tamu.7B.16* | 7B | 15.6 | YLD | 12CH, AcrossME3, ME3, Across ME1234 | 7B1 | 24 | 21.5–24.5 | 6.0–8.4 | 2.4–4.3 | 4.2–6.0 | 3.3–10.9 | 2.5–4.9 | 1.8–2.0 | −1.87-(−7.23) | TAM 111 | A/C | T/C | HD | *Yang et al. (2020b)* |
| *Qyld.tamu.7D.61* | 7D | 60.6 | YLD | 12UVD, across all env, AcrossME3 | 7D | 79 | 78.5–80.5 | 3.5–22.1 | 1.4–11.2 | 1.6–20.7 | 9.7–20.1 | 1.6–13.3 | 0.5–18.5 | 1.24–8.73 | TAM 112 | T/C | C/T | TW, HT | *Juliana et al. (2019)* |
| *Qyld.tamu.7D.64* | 7D | 64.3 | YLD | 12CH, 17BD, ME3 | 7D | 80 | 77.5–80.5 | 3.9–6.4 | | | 2.5–12.5 | | | 4.28–6.45 | TAM 112 | C/T | T/C | HT | *Cabral et al. (2018)* |
| *Qyld.tamu.7D.84* | 7D | 84.3 | YLD | 17BI, across all env, AcrossME1 | 7D | 97 | 95.5–98.5 | 5.0–17.0 | 0.0–6.6 | 1.0–17.0 | 11.1–28.9 | 0.0–5.8 | 5.5–28.9 | −0.004-(−21.80) | TAM 111 | G/A | A/G | | |
| *Qyld.tamu.7D.591* | 7D | 591.2 | YLD | 12BD, AcrossME2 | 7D | 181 | 180.5–181.5 | 4.7–7.0 | 2.0 | 5.0 | 7.8–13.5 | 3.5 | 4.4 | 2.47–3.00 | TAM 112 | A/G | C/T | | *Yang et al. (2020b)* and *Tan et al. (2017)* |
| *Qtw.tamu.1A.12* | 1A | 11.8 | TW | across all env | 1A | 9 | 8.5–9.5 | 12.6 | 7.8 | 4.8 | 1.6 | 1.0 | 0.6 | −1.87 | TAM 111 | A/G | A/G | y | *Guo et al. (2020)* |
| *Qtw.tamu.1A.381* | 1A | 380.7 | TW | across all env, AcrossME2 | 1A | 51 | 50.5–51.5 | 9.0–14.7 | 6.1–10.6 | 2.9–4.1 | 1.9–2.3 | 1.4–1.5 | 0.5–0.8 | −2.06-(−2.18) | TAM 111 | C/G | G/A | | |
| *Qtw.tamu.1A.485* | 1A | 485.2 | TW | 13EP5, across all env, AcrossME2 | 1A | 64 | 63.5–64.5 | 4.1–17.4 | 7.4–11.5 | 4.3–5.8 | 2.1–5.2 | 1.5–1.8 | 0.6–0.8 | −2.25-(−4.91) | TAM 111 | T/C | A/C | | *Dhakal et al. (2021)* |
| *Qtw.tamu.1D.375* | 1D | 375.4 | TW | 12EP3, across all env | 1D | 46 | 45.5–46.5 | 4.9–16.9 | 9.6 | 7.2 | 2.2–11.9 | 1.3 | 0.9 | 2.07–6.13 | TAM 112 | A/C | A/G | | *Jin et al. (2016)* |
| *Qtw.tamu.1D.422* | 1D | 421.8 | TW | 12CH, across all env, AcrossME2, ME2, Across ME12 | 1D | 69 | 68.5–69.5 | 3.5–20.1 | 5.1–11.4 | 0.1–9.4 | 2.7–11.0 | 1.4–10.2 | 0.2–1.3 | 2.19–6.41 | TAM 112 | T/C | C/A | y | |
| *Qtw.tamu.2B.709* | 2B | 708.7 | TW | 13EP5, across all env, AcrossME2 | 2B | 124 | 123.5–124.5 | 7.3–20.3 | 11.3–12.3 | 6.0–8.1 | 2.2–10.1 | 1.6–2.8 | 0.6–0.7 | 2.36–6.91 | TAM 111 | T/G | G/A | | |
| *Qtw.tamu.2D.487* | 2D | 486.8 | TW | across all env, AcrossME2 | 2D | 104 | 102.5–104.5 | 13.8–19.4 | 8.9–12.2 | 4.9–7.3 | 2.2–2.9 | 1.6–2.2 | 0.6–0.7 | 2.33–2.47 | TAM 112 | G/C | C/A | | *Dhakal et al. (2021)* |

Dhakal et al. (2021), *PeerJ*, DOI 10.7717/peerj.12350

**Table 3** (*continued*)

| QTL (underlined involved interactions) | Chr | Position[a] (Mb) | Trait[b] | Environment[c] | Linkage[d] | Peak[e] (cM) | QTL CI[f] (cM) | LOD[g] | LOD (A) | LOD (A*E) | PVE[h] (%) | PVE (A) (%) | PVE (A*E) (%) | ADD[i] | SNP alleles increase traits | Left SNPs alleles[j] | Right SNPs alleles[k] | Pleiotropic QTL | Citation for known QTL |
|---|---|---|---|---|---|---|---|---|---|---|---|---|---|---|---|---|---|---|---|
| *Qtw.tamu.3B.507* | 3B | 507.0 | TW | 12CH, across all env, AcrossME2 | 3B | 27 | 24.5–27.5 | 4.6–13.8 | 8.2–8.3 | 3.9–5.5 | 1.6–6.6 | 1.1–2.0 | 0.6–0.8 | −1.92-(−5.24) | TAM 111 | G/C | T/C | | |
| *Qtw.tamu.3D.549* | 3D | 548.6 | TW | across all env, AcrossME2 | 3D | 53 | 49.5–53.5 | 9.3–15.6 | 7.1–12.9 | 2.2–2.8 | 2.0–2.1 | 1.67–1.74 | 0.3–0.4 | −2.21-(−2.39) | TAM 111 | A/G | T/C | | |
| *Qtw.tamu.3D.555* | 3D | 554.7 | TW | across all env, AcrossME2 | 3D | 58 | 57.5–58.5 | 11.0–16.6 | 9.4–14.6 | 1.6–2.0 | 2.3–2.7 | 1.9–2.3 | 0.37–0.43 | −2.55-(−2.57) | TAM 111 | C/T | A/C | | |
| *Qtw.tamu.3D.563* | 3D | 562.7 | TW | across all env, AcrossME2 | 3D | 63 | 62.5–63.5 | 10.6–14.0 | 9.7–12.3 | 1.0–1.8 | 2.0–2.9 | 1.6–2.3 | 0.4–0.6 | −2.35-(−2.57) | TAM 111 | T/C | T/G | | |
| *Qtw.tamu.4D.29* | 4D | 29.0 | TW | 12UV7, AcrossME1 | 4D | 6 | 4.5–8.5 | 5.6–7.1 | 3.0 | 4.1 | 2.5–5.0 | 2.0 | 3.1 | 1.95–7.66 | TAM 112 | A/G | G/T | | *Dhakal et al. (2021)* and *Yang et al. (2020b)* |
| Qtw.tamu.4D.63 | 4D | 62.8 | TW | 12BD, 12EP1, 12EP3, 13EP2, 14CH, across all env, AcrossME2 | 4D | 11 | 10.5–11.5 | 3.9–35.6 | 14.6–18.6 | 10.4–17.0 | 5.6–16.3 | 2.4–3.6 | 3.2–4.3 | 2.89–10.35 | TAM 112 | C/T | C/T | | |
| *Qtw.tamu.4D.110* | 4D | 109.8 | TW | 12UV5, 13EP3, across all env, AcrossME1 | 4D | 13 | 12.5–13.5 | 3.5–14.9 | 4.8–5.7 | 5.8–10.0 | 2.4–19.4 | 0.6–3.8 | 1.8–5.5 | 1.47–8.01 | TAM 112 | C/T | T/C | y | *Dhakal et al. (2021)* and *Yang et al. (2020b)* |
| *Qtw.tamu.5A.74* | 5A | 73.8 | TW | across all env, AcrossME2 | 5A | 40 | 39.5–40.5 | 10.1–14.4 | 6.0–9.4 | 4.1–5.0 | 1.6–2.2 | 1.2–1.5 | 0.4–0.7 | 2.02–2.05 | TAM 112 | T/C | G/A | | |
| *Qtw.tamu.5A.157* | 5A | 157.3 | TW | across all env, AcrossME2 | 5A | 43 | 42.5–43.5 | 10.1–13.5 | 6.0–8.7 | 4.1–4.8 | 1.5–2.2 | 1.1–1.5 | 0.4–0.7 | 1.97–2.03 | TAM 112 | G/C | G/A | | |
| Qtw.tamu.5A.702 | 5A | 702.0 | TW | across all env, AcrossME1 | 5A | 194 | 192.5–194.5 | 6.8–14.5 | 5.5–11.1 | 1.3–3.4 | 1.7–4.0 | 1.4–3.6 | 0.2–0.4 | 2.22–2.66 | TAM 112 | A/G | T/C | | |
| *Qtw.tamu.5B.589* | 5B | 589.4 | TW | across all env, AcrossME1, AcrossME2 | 5B | 75 | 74.5–75.5 | 6.8–16.6 | 5.8–13.4 | 1.0–3.2 | 2.3–4.4 | 1.7–3.8 | 0.6–0.8 | −2.28-(−2.71) | TAM 111 | A/G | C/T | | *Zhang et al. (2018)* |
| Qtw.tamu.5B.646 | 5B | 646.0 | TW | across all env, AcrossME2 | 5B | 94 | 90.5–94.5 | 9.9–14.4 | 8.2–11.2 | 1.7–3.2 | 2.0–2.7 | 1.4–2.0 | 0.5–0.7 | −2.23-(−2.36) | TAM 111 | A/G | T/C | | |
| *Qtw.tamu.6A.7* | 6A | 7.2 | TW | across all env, AcrossME2 | 6A | 12 | 11.5–12.5 | 9.2–15.0 | 6.0–11.4 | 3.2–3.5 | 2.1–2.2 | 1.47–1.49 | 0.6–0.8 | 2.03–2.26 | TAM 112 | T/C | T/C | | |
| Qtw.tamu.6A.12 | 6A | 12.4 | TW | 12BD, across all env, AcrossME1 | 6A | 20 | 19.5–20.5 | 4.1–17.9 | 9.5–12.1 | 2.2–5.8 | 2.0–8.5 | 1.6–6.2 | 0.2–0.4 | 2.32–3.58 | TAM 112 | G/C | T/G | y | |
| *Qtw.tamu.6A.603* | 6A | 603.3 | TW | 12UV7, across all env, AcrossME1 | 6A | 134 | 133.5–134.5 | 20.0–24.4 | 0.0–1.6 | 20.3–24.4 | 4.8–21.4 | 0.0–1.1 | 4.8–20.3 | −0.08–16.58 | TAM 111, TAM 112 | A/G | T/C | | *Guo et al. (2020)*; *Yang et al. (2020a)* |
| *Qtw.tamu.6A.609* | 6A | 608.5 | TW | 13EP5, across all env, AcrossME2 | 6A | 139 | 138.5–139.5 | 4.6–14.0 | 6.7–8.6 | 3.9–5.3 | 1.7–5.8 | 1.1–1.6 | 0.6–0.8 | −5.19-(−1.96) | TAM 111 | T/C | T/C | y | *Guo et al. (2020)*; *Yang et al. (2020a)* |
| *Qtw.tamu.6A.612* | 6A | 611.6 | TW | 12CH, across all env, AcrossME2 | 6A | 143 | 142.5–143.5 | 6.6–14.4 | 5.9–7.7 | 5.5–6.8 | 1.8–10.1 | 1.0–1.4 | 0.8–1.2 | −1.85-(−6.47) | TAM 111 | A/G | T/C | | |
| Qtw.tamu.6B.130 | 6B | 130.3 | TW | 12UVL, 14EP4, across all env, AcrossME1 | 6B2 | 1 | 0–2.5 | 5.0–15.1 | 5.8–6.1 | 5.6–9.3 | 2.4–17.6 | 0.8–4.2 | 1.6–4.5 | 1.64–7.88 | TAM 112 | T/G | A/C | | *Juliana et al. (2019)* |
| *Qtw.tamu.6B.466* | 6B | 466.0 | TW | 11EP5, 12BD, 13EP2, 14CH, 17BD, across all env, AcrossME1, ME1, Across ME12 | 6B2 | 7 | 6.5–7.5 | 3.8–23.9 | 3.4–11.4 | 2.4–12.5 | 3.4–15.4 | 1.5–6.8 | 2.0–6.4 | 2.28–7.77 | TAM 112 | G/A | A/G | | |
| *Qtw.tamu.6B.559* | 6B | 559.4 | TW | 13EP5, AcrossME2, ME2, Across ME12 | 6B2 | 8 | 7.5–8.5 | 4.1–15.2 | 2.4–5.0 | 1.7–10.2 | 3.7–15.4 | 1.2–4.8 | 2.5–3.4 | 1.86–8.55 | TAM 112 | A/G | T/C | | *Juliana et al. (2019)* |
| *Qtw.tamu.6D.459* | 6D | 459.2 | TW | 12UV5, across all env, across all env, AcrossME1, AcrossME2, ME1 | 6D | 99 | 98.5–99.5 | 4.0–21.3 | 10.1–15.8 | 1.8–6.1 | 2.1–11.3 | 1.4–6.9 | 0.5–2.7 | −2.21-(−7.23) | TAM 111 | A/G | A/G | | *Yang et al. (2020b)* |
| Qtw.tamu.7B.9 | 7B | 8.5 | TW | 12CH, across all env, AcrossME2 | 7B1 | 19 | 17.5–21.5 | 3.5–17.3 | 7.5–9.5 | 5.5–9.8 | 1.9–5.4 | 1.0–2.2 | 0.8–0.9 | −1.86-(−4.78) | TAM 111 | A/C | A/C | | *Juliana et al. (2019)* |
| *Qtw.tamu.7B.64* | 7B | 64.5 | TW | across all env, AcrossME2 | 7B1 | 48 | 47.5–48.5 | 9.2–13.0 | 7.2–9.5 | 2.1–3.6 | 1.7–2.3 | 1.2–1.7 | 0.46–0.55 | −2.05-(−2.21) | TAM 111 | G/A | C/T | | |
| *Qtw.tamu.7D.61* | 7D | 60.6 | TW | 11EP5, 12CH, across all env, AcrossME2 | 7D | 79 | 78.5–79.5 | 6.9–27.0 | 11.2–12.4 | 12.4–15.8 | 3.3–21.6 | 1.5–3.1 | 1.8–2.8 | 2.25–8.48 | TAM 112 | T/C | C/T | y | |
| *Qtw.tamu.7D.604* | 7D | 604.0 | TW | 13EP5, across all env, AcrossME2 | 7D | 206 | 204.5–206.5 | 5.2–15.3 | 5.3–8.8 | 5.4–6.4 | 1.7–6.8 | 1.1–1.3 | 0.5–0.8 | −1.91-(−5.61) | TAM 111 | | G/A | | |
| *Qht.tamu.4D.21* | 4D | 20.6 | HT | 12BD, DRY, across all env | 4D | 0 | 0–0.5 | 3.7–8.8 | 1.2 | 7.6 | 4.9–6.7 | 1.2 | 3.7 | 0.21–0.47 | TAM 112 | A/G | T/C | y | |

**Table 3** (*continued*)

| QTL (underlined involved interactions) | Chr | Position[a] (Mb) | Trait[b] | Environment[c] | Linkage[d] | Peak[e] (cM) | QTL CI[f] (cM) | LOD[g] | LOD (A) | LOD (A*E) | PVE[h] (%) | PVE (A) (%) | PVE (A*E) (%) | ADD[i] | SNP alleles increase traits | Left SNPs alleles[j] | Right SNPs alleles[k] | Pleiotropic QTL | Citation for known QTL |
|---|---|---|---|---|---|---|---|---|---|---|---|---|---|---|---|---|---|---|---|
| *Qht.tamu.7D.61* | 7D | 60.6 | HT | 17EI | 7D | 79 | 77.5–80.5 | 3.3 | | | 8.1 | | | −1.05 | TAM 111 | T/C | C/T | y | |
| Qht.tamu.7D.64 | 7D | 64.3 | HT | 11EP5, across all env | 7D | 80 | 78.5–80.5 | 3.8–13.2 | 2.0 | 5.0 | 13.5–19.6 | 3.5 | 4.4 | −0.41-(−1.63) | TAM 111 | T/C | C/T | y | |
| *Qhd.tamu.1A.12* | 1A | 11.8 | HD | across all env | 1A | 9 | 7.5–9.5 | 8.6 | 5.6 | 2.9 | 3.1 | 2.1 | 1.0 | 0.28 | TAM 112 | A/G | A/G | y | |
| *Qhd.tamu.4B.659* | 4B | 659.2 | HD | 11EP2, across all env | 4B | 94 | 93.5–94.5 | 6.7–12.4 | 7.9 | 4.5 | 7.2–18.3 | 3.0 | 4.3 | 0.34–1.53 | TAM 112 | A/G | A/G | y | |
| *Qhd.tamu.7B.16* | 7B | 15.6 | HD | across all env | 7B1 | 24 | 21.5–24.5 | 8.7 | 4.4 | 4.4 | 2.2 | 1.6 | 0.6 | 0.25 | TAM 112 | A/C | T/C | y | |

**Notes.**

[a] Physical position based on IWGSC RefSeq v 1.0 Mega base pair position.

[b] Trait abbreviations: grain yield (YLD), test weight (TW), days to heading (HD), and plant height (HT).

[c] Abbreviations: 11, Year 2011; 12, Year 2012; 13, Year 2013; 14, Year 2014; 17, Year 2017; BD, Bushland dry (I0), TX; BI, Bushland Irrigated ($I_{100}$), TX; CH, Chillicothe (I0), TX; CVI, Clovis Irrigated ($I_{100}$), NM; EP1, Etter (I0), TX; EP2, Etter (I50), TX; EP3, Etter (I65), TX; EP4, Etter (I75), TX; EP5, Etter (100), TX; UVD, Uvalde dry (I0), TX; UV5, Uvalde (I50), TX; UV7, Uvalde (I70), TX; UVL, Uvalde ($I_{100}$), TX. Ind Env-ADD, single environment QTL analysis; Multi Env-ADD, single trait multiple environment QTL analysis; Ind ME-ADD, Individual mega environment QTL analysis; Across ME-ADD, single trait multiple mega environment QTL analysis.

Environment 1 to 28 for yield are: 11BD, 11CH, 11EP1, 11EP2, 11EP3, 11EP4, 11EP5, 12BD, 12CH, 12EP1, 12EP2, 12EP3, 12UV5, 12UV7, 12UVL, 12UVD, 13EP2, 13EP3, 13EP4, 13EP5, 13UVL, 14CH, 14EP4, 14EP5, 17BD, 17BI, 17CVI, and 17EI.

Environment 1 to 19 for test weight are: 11EP5, 12BD, 12CH, 12EP1, 12EP2, 12EP3, 12UV7, 12UVL, 12UV5, 13EP2, 13EP3, 13EP4, 13EP5, 14CH, 14EP4, 14EP5, 17BD, 17BI, and 17EI.

Environment 1 to 11 for height are: 11BD, 11EP1, 11EP2, 11EP3, 11EP4, 11EP5, 12BD, 12CH,1 7BD, 17CVI, and 17EI.

Environment 1 to 11 for heading date are: 11BD, 11EP1, 11EP2, 11EP3, 11EP4, 11EP5, 12BD, 12CH, 17BD, 17CVI, and 17EI.

Mega-environments for traits: YLD had ME1 (11CH, 14EP4, 14EP5, 17BI, and 17CVI), ME2 (11BD, 11EP5, 12BD,12EP1, 12EP2, and 17EI), ME3 (12CH, 12UVD, 13UVL, and 17BD), and ME4 (12UV5, 12UV7, 12UVL, and 14CH); TW had ME1 (12BD, 12EP2, 12UV7, 12UVL, 12UV5, 13EP3, and 14CH) and ME2 (11EP5, 12CH, 12EP1, 12EP3, 13EP2, 13EP4, 13EP5, 14EP4, 14EP5, 17BD, 17BI, and 17EI). HT had ME1 (11EP3, 11BD, 12BD, 17CVI, 12CH), ME2 (11EP1, 11EP2, 11EP4, 11EP5, 17BD, 17EI). HD had ME1 (12EP2, 11EP4, 11BD, 12EP1, 17BD) and ME2 (12EP3, 12BD, 11EP5, 11EP1, 11EP3).

[d] Linkage group.

[e] Centi Morgan distance.

[f] QTL 95% confidence interval.

[g] Logarithm of odds, A, LOD due to A; A*E, LOD due to A*E interaction.

[h] Phenotypic variance explained, A, PVE explained by Additive effect, A*E, PVE by additive-by-environment interaction effect.

[i] Additive effects of the QTL. Positive value indicates the favorable allele came from female parent TAM 112 and negative value indicates the favorable allele came from male parent TAM 111.

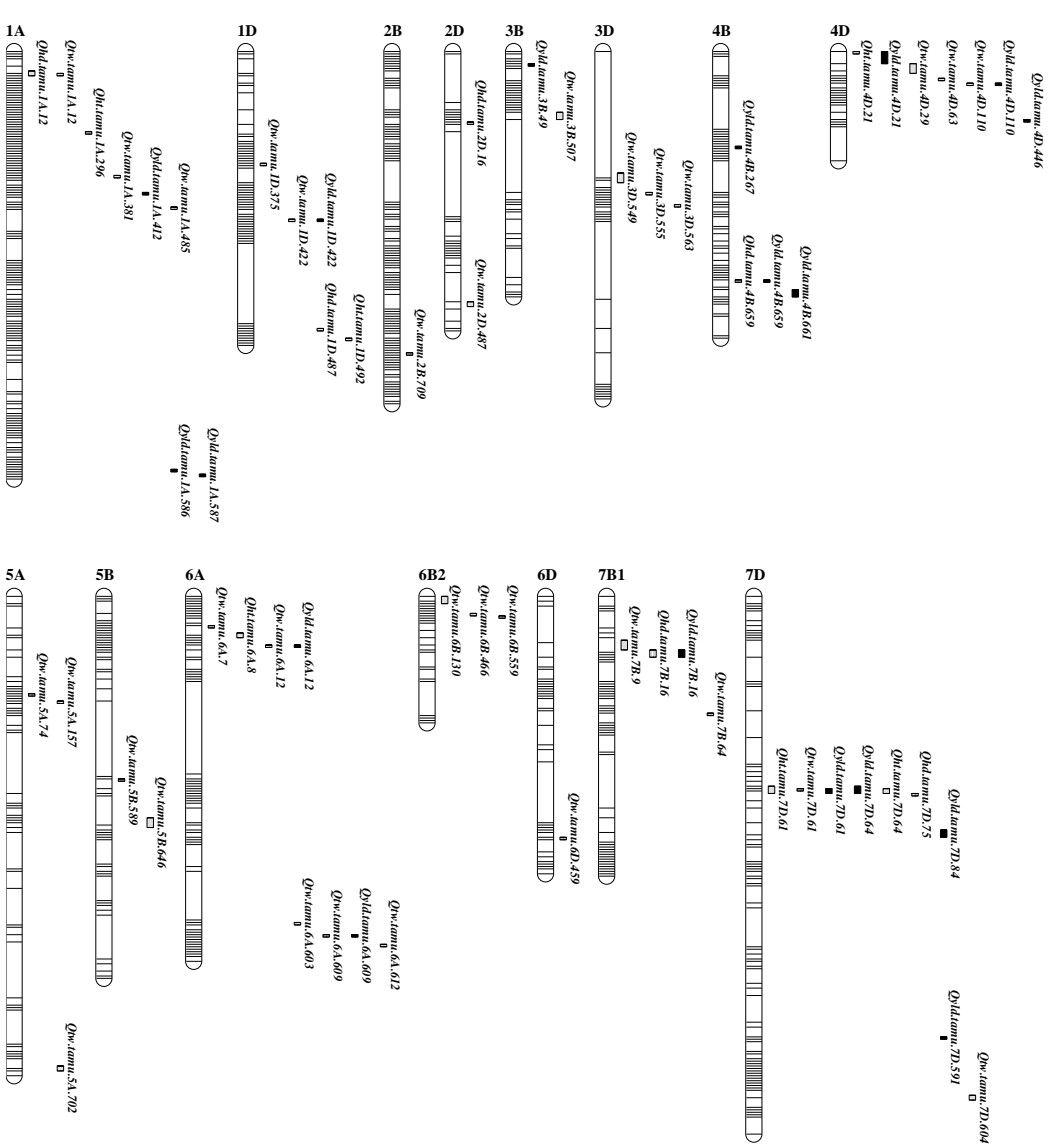

**Figure 2  Genetic maps showing the positions of QTL for grain yield and agronomic traits from QTL analysis in TAM 112 × TAM 111 RIL population.** Markers are represented by horizontal stripes inside a linkage group. Traits include grain yield (YLD), test weight (TW), plant height (HT), and heading date (HD). Identified QTL were designated in the format as *Qtrait.tamu.chrom.Mb*. The bar length is the flanking marker intervals in cM.

improvement. Two QTL, *Qyld.tamu.1A.587* and *Qyld.tamu.7D.61*, increased yield by 4.43 and 6.57 g m$^{-2}$ with TAM 112 allele contributing 14.6% and 13.3% additive effect for percentage of variations explained (PVE) across the four mega-environments and ME3 (including 12CH, 12UVD, 13UVL, and 17BD) analyses, respectively (Table S4). A third QTL at 659.2 Mb on 4B increased yield by 6.43 g m$^{-2}$ with TAM 112 allele contributing 4.9% additive PVE. A set of four additional QTL, at 20.6 and 109.8 Mb on 4D, 15.6 Mb on 7B, and 84.3 Mb on 7D, increased yield by 2.55 to 8.16 g m$^{-2}$ with TAM 111 alleles

contributing additive PVE ranging from 4.9% to 7.9%. These seven yield QTL will be our focus for further application.

The additive-by-environment interaction increased yield by up to 37.83 g m$^{-2}$ from *Qyld.tamu.4B.267* under 17CVI with TAM 111 allele while the same QTL had favorable allele from TAM 112 that only increased 11.13 g m$^{-2}$ from interaction under 17BI when it was analyzed for across all 28 environments. Furthermore, the same QTL had interactional effect that increased yield by 34.03 g m$^{-2}$ under 17CVI with TAM 111 allele and increased yield by 14.92 g m$^{-2}$ under 17BI with TAM 112 allele from analyses of across ME1 including 11CH, 14EP4, 14EP5, 17BI, and 17CVI. Similarly, *Qyld.tamu.7D.84* had interactional effect increased yield by 20.3 g m$^{-2}$ under 17BI with TAM 111 allele and increased by 11.12 g m$^{-2}$ under 12UV5 with TAM 112 allele when analyzed across all 28 environments. The same QTL had an interactional effect of increasing yield by 13.53 g m$^{-2}$ with TAM 111 allele under 17BI when it was analyzed across ME1 including 11CH, 14EP4, 14EP5, 17BI, and 17CVI.

Five major QTL had QTL-by-environment interactions from TAM 111 alleles. *Qyld.tamu.3B.49* increased yield by 10.8 g m$^{-2}$ under 12UVL when it was analyzed across ME4 including 12UV5, 12UV7, 12UVL, and 14CH. The effect of increasing yield by 11.01 and 26.24 g m$^{-2}$ were observed when all 28 environments were analyzed for QTL *Qyld.tamu.4D.446* under 11EP4, and *Qyld.tamu.4D.110* under 17BI, respectively. The two QTL, *Qyld.tamu.7D.84* and *Qyld.tamu.4D.110* had interactional effect of increasing yield by 13.53 and 19.75 g m$^{-2}$ with TAM 111 allele under 17BI from across ME1 analyses (Table S4).

## Test weight

A total of 32 unique QTL for test weight were identified. Six consistent and pleiotropic QTL were mapped on chromosomes 1A at 11.8 Mb, 1D at 421.8 Mb, 4D at 109.8 Mb, 6A at 12.4 Mb and 608.5 Mb, and 7D at 60.6 Mb. Additional 26 consistent QTL were located on 14 chromosomes (Table 3, Fig. 2, and Fig. S2). The favorable alleles for TW were from TAM 111 on chromosome 1A at 11.8, 380.7, and 485.2 Mb, 3B at 507.0 Mb, 3D at 548.6, 554.7, and 562.7 Mb, 5B at 589.4 and 646.0 Mb, 6A at 603.3, 608.5 and 611.6 Mb, 6D at 459.2 Mb, 7B at 8.5 and 64.5 Mb, and 7D at 604 Mb that explained up to 11.3% phenotypic variation and increased TW by 7.23 kg m$^{-3}$ with *Qtw.tamu.6D.459* in 12UV5 (Table 3, Table S4). The rest of the QTL had favorable allele from TAM 112 that increased TW by up to 10.35 kg m$^{-3}$ with *Qtw.tamu.4D.63* in 13EP2. The QTL on the chromosome 6A at 603.3 Mb received favorable alleles from both parents depending on the particular environments. *Qtw.tamu.6A.603* explained up to 21.4% of the phenotypic variation and increased TW by 16.58 kg m$^{-3}$ with alleles from TAM 112 in 12UV7 but it only increased 0.08 kg m$^{-3}$ with TAM 111 allele in the analyses across all 19 environments.

Based on PVE of additive effects, only *Qtw.tamu.1D.422* along with other four QTL on 4D at 109.8 Mb, 6A at 12.4 Mb, and on 6B at 466.0 and 559.4 Mb, had values ranging from 3.8% to 10.2% with TAM 112 alleles that increased test weight by 2.26 to 3.47 kg m$^{-3}$ (Table S4). One QTL *Qtw.tamu.6D.459* increased test weight by 3.66 kg m$^{-3}$ with TAM 111 allele.

For the QTL-by-environmental interactions based on across all individual environment and across ME2 analyses, *Qtw.tamu.4D.63* had additive effects of 7.34 and 7.05 kg m$^{-3}$ in environment 13EP2, respectively while their QTL additive effects only increased 2.89 and 3.18 kg/m3 (Table S4). *Qtw.tamu.6A.603* had interactional effect of increasing TW by 16.65 and 15.14 kg m$^{-3}$ under the environment 12UV7 from the analyses of across all environments and ME1, respectively (Table S2). All these favorable alleles were from TAM 112.

## Pleiotropic QTL to grain yield and test weight

From the multiple trait QTL analyses, nine unique consistent QTL regions for grain yield are also linked to agronomic traits (Table 3, Table S4, and Fig. 2). They were detected on chromosome 1A, 1D, 4B, 4D, 6A, 7B, and 7D. QTL linked to HD and TW on chromosomes 1A at 11.8 Mb increased HD by 0.28 day with TAM 112 allele while the TAM 111 alleles increased TW by 1.87 kg m$^{-3}$ (Table 3, Table S4). QTL on 1D at 421.8 Mb had a favorable allele from TAM 112 that increased TW by 6.41 kg m$^{-3}$ and YLD by 6.87 g m$^{-2}$. QTL on chromosome 4B at 659.2 Mb was associated with HD and YLD. Allele from TAM 112 increased HD by 1.53 days and YLD by 9.88 g m$^{-2}$. Plant height and YLD were associated with QTL on chromosome 4D at 20.6 Mb. TAM 112 allele increased HT by 0.47 cm under combined DRY while TAM 111 allele increased YLD by 17.32 g m$^{-2}$ under 17CVI. Another QTL on chromosome 4D at 109.8 Mb was associated with TW and YLD. TAM 112 allele increased TW by up to 8.01 kg m$^{-3}$ in 13EP3 while TAM 111 allele increased YLD by 28.06 g m$^{-2}$ under 17BI. The two QTL on chromosome 6A at 12.4 Mb and 608.5 Mb were associated with both TW and YLD. TAM 112 allele of QTL at 12.4 Mb increased TW by 3.58 kg m$^{-3}$ while TAM 111 allele increased YLD by 19.58 g m$^{-2}$. For the QTL at 608.5 Mb, TAM 111 allele increased TW by 5.19 kg m$^{-3}$ under 13EP5 and YLD by 3.18 g m$^{-2}$ under ME3. Heading date and YLD were co-located on chromosome 7B at 15.6 Mb. TAM 112 allele increased HD by 0.25 day while TAM 111 allele increased YLD by up to 7.23 g m$^{-2}$ under 12CH. QTL on chromosome 7D at 60.6 Mb was associated with HT, TW, and YLD. The favorable allele from TAM 111 increased HT by 1.05 cm under 17EI while the favorable allele from TAM 112 increased TW and YLD by up to 8.48 kg m$^{-3}$ under 11EP5 and 8.73 g m$^{-2}$ under 12UVD, respectively. Another QTL on chromosome 7D at 64.3 Mb was associated with HT and yield. TAM 111 allele increased HT by 1.63 cm under 11EP5 and TAM 112 allele increased yield by 6.45 g m$^{-2}$ under 12CH. Among the five pleiotropic QTL associated with both TW and yield, the two QTL on 1D and 7D have favorable alleles increasing both traits from TAM 112 while the QTL on 4D at 109.8 and 6A at 12.4 Mb had TAM 112 allele for TW and TAM 111 allele for yield. The QTL on 6A at 608.5 Mb had TAM 111 allele increasing both traits (Table 3, Table S4).

## Interactions of epistasis, epistasis-by-environment, and additive-by-environment

There were 359 interactions of additive-by-additive, additive-by-environment, and epistasis-by-environment with a total LOD $\geq$ 12 for all traits (Table S5, Fig. S4). Only 139 out of 359 interactions had LOD(AA) >10.0. There were only two interactions for HD,

one for yield, and the rest of the 136 for TW. None of the interactions was the same as any of those major consistent and pleiotropic QTL for heading date and yield while there were five major consistent QTL associated with the epistasis interactions for TW (Table S5). They were a marker linked to *Qtw.tamu.5A.702* interacting with two other markers on 1A at 465.5 Mb and on 5A at 584.4 Mb, a marker linked to *Qtw.tamu.5B.646* interacting with IWB5813 on 2B at 25.2 Mb, a marker linked to *Qtw.tamu.6A.12* interacting with a marker on 1B at 466 Mb, IWB38972 linked to *Qtw.tamu.6B.130* interacting with a marker on 1A at 544.6 Mb, IWB6455 linked to *Qtw.tamu.7B.9* interacting with a marker on 1D at 418.5 Mb. However, they only explained TW variations by 1.3% to 2.8% with additive effects from epistasis less than 0.25 kg m$^{-3}$. Furthermore, neither of any interactions from epistasis, the interactions between either marker from the epistasis with the environment, or the epistasis-by-environment interactions had effects that increased TW by more than 1 kg m$^{-3}$.

For the interaction of yield between a marker on 5B at 655.5 Mb and a marker on 7B at 740.1 Mb, its epistasis increased yield by 33.74 g m$^{-2}$. The marker on 5B at 655.5 Mb increased yield by more than 30 g m$^{-2}$ with TAM 111 allele under drier environments 11CH, 12EP3, and 12UV5, and with TAM 112 allele under higher irrigated environments 11EP5, 12UV7, and 17BI. The marker on 7B at 740.1 Mb increased yield by more than 30 g m$^{-2}$ with TAM 111 alleles under environments 11EP5, 12UV5, and 13UVL, and with TAM 112 alleles under environments 12CH and 17EI. The epistasis-by-environment interaction effects increased yield by more than 30 g m$^{-2}$ with TAM 111 alleles under less irrigated environments 11EP2, 12UV7, 13EP2, and 13EP3, but with TAM 112 alleles under highly irrigated environments 11EP3, 11EP4, 11EP5, 17BI and 17CVI (Table S5). Neither marker regions were involved with major QTL for yield.

Among the other 220 interactions with LOD(AA) <10.0, six interactions for HD, eight interactions for HT, 53 interactions for TW, and 153 interactions for yield. For TW, there were five interactions that increased TW by more than 0.3 kg m$^{-3}$ with TAM 111 alleles and 14 interactions with TAM 112 alleles but all of them had effect less than 0.5 kg m$^{-3}$. One major QTL *Qtw.tamu.4D.63* interacting with 12EP1, 12EP3, 17BD increased TW by 0.38–0.44 kg m$^{-3}$ with TAM 112 allele while the other two major QTL *Qtw.tamu.2D.487* and *Qtw.tamu.7B.9* had interactional effects less than 0.3 kg m$^{-3}$ (Table S4).

For yield, a total of 1,092 interactions increased yield by more than 50 g m$^{-2}$ based on the 153 pairs of QTL across 28 environments with 513 from TAM 112 alleles and 579 from TAM 111 alleles. A subset of 87 interactions increased yield by more than 100 g m$^{-2}$. At the first locus of the epistasis, environments 11EP4, 11EP5, 12UV5, and 17BI interacted with a set of 40, 55, 43, and 41 markers increased YLD more than 50 g m$^{-2}$ while the corresponding subsets of four, three, three, and one marker increased yield by more than 100 g m$^{-2}$. At the second locus, environments 11EP4, 11EP5, 12UV7, 12UVL, 17BI interacted with a set of 35, 40, 29, 17, and 23 markers increased yield by more than 50 g m$^{-2}$ while the corresponding subsets of one, five, two, three, and two markers increased yield by more than 100 g m$^{-2}$. For epistasis-by-environment interactions, environments 11EP3, 11EP4, 11EP5, 12UV5, 12UV7, 12UVL, 17BI, and 17CVI interacted with a set of 38, 52, 55, 75, 46, 45, 75, and 29 epistasis increased yield by more than 50 g m$^{-2}$ while the corresponding subset of four,

seven, 14, 14, two, three, 18, and one marker increased yield by more than 100 g m$^{-2}$ with three interactions. The top four marker pairs increased yield by more than 100 g m$^{-2}$ with additive-by-environment or epistasis-by-environment interactions. A marker on 6A at 613.8 Mb interacted with 12UV5 increased yield by 127.2 g m$^{-2}$ with TAM 111 allele. The epistasis between 6A marker with a marker on 7D at 585.6 Mb increased yield by 105.7 g m$^{-2}$ under 12UV5 with TAM 112 allele while it increased yield by 106.2 g m$^{-2}$ under 11EP4 with TAM 111 alleles. The 2nd set of three interactions were as follow: a marker on 4A at 378.3 Mb and IWA5751 on 4D at 408.8 Mb interacted with 11EP4 increased yield by 104.19 g m$^{-2}$ with TAM 112 allele and by 110.72 g m$^{-2}$ with TAM 111 allele, respectively; the epistasis between these two markers interacted with 12UV5 increased yield by 102.07 g m$^{-2}$ with TAM 111 allele. The 3rd set of three interactions were: marker IWB52359 on 7D at 40.1 Mb interacted with 12UVL and 12UV7 increased yield by 128.98 and 109.82 g m$^{-2}$ with TAM 112 allele, respectively; its epistasis with a marker on 7B at 6.8 Mb interacting with 17BI increased yield by 146.04 g m$^{-2}$ with TAM 111 allele. The fourth set of three interactions were all epistasis-by-environment effects between IWB73713 on 1B at 675.6 Mb and IWA1924 on 6D at 461.4 Mb that increased yield by 105.73, 106.52, and 106.25 g m$^{-2}$, respectively, under 11EP5, 12UV5, and 12UVL with the first interactional allele from TAM 111 and the rest two from TAM 112 (Table S5). The highest effect from TAM 111 allele increased yield by 184.15 g m$^{-2}$ that was from a major QTL *Qyld.tamu.6A.12* under 17BI while the highest effect from TAM 112 alleles increased yield by 155.18 g m$^{-2}$ was from epistasis-by-environment effect between IWA4746 on 2D at 14.4 Mb and a marker on 3A at 7.6 Mb under 11EP5. Two other major QTL, *Qyld.tamu.4D.21* and *Qyld.tamu.4B.659*, were involved with epistasis-by-environment interactions but most interactional effect only increased yield by less than 100 g m$^{-2}$ (Table S5).

## DISCUSSION

Highly heritable traits are important to breeders. The yield and agronomic traits analyzed in this study were moderate to highly heritable (Table 2). The genotypic variances were larger than the genotype-by-environment and residual variance for HD and TW. Higher heritability indicated that these traits were largely genetically controlled, making them suitable for genetic gain from selection in a breeding program. Higher heritability in yield and agronomic traits have been reported by *Zhang et al. (2018)*. Since yield is controlled by many genes with each showed minor effects and is easily influenced by the environment, it was unusual to see high heritability for yield approach 0.7 (*Li et al., 2007*). However, highly significant G × E interactions were found for all the traits. Given all the possible environmental conditions of this study, observing significant environmental and genotype-by-environment interaction variances are expected. This population was planted in a wide range of environments, including diverse soil types, precipitations, and temperatures. Our testing environment included locations with day temperature >30 °C in Southern Texas to locations with day temperature <10 °C in the High Plains of Texas. These two environmental covariables significantly alter genotype expression across environments in wheat yield (*Kuchel et al., 2007*; *Saini & Aspinall, 1982*). All the dryland experiments

in the High Plains of Texas received less than five inches of rainfall during the growing seasons, which is typical in this region. This population suffered an extreme drought in 2011 and freeze damage in the late growing season in 2013. Globally wheat benefitted from reduced height as that increased harvest index, straw strength, and yield. It is also known that reduced height, accompanied by a higher input level, imparted a significant increase in yield (*Borlaug, 1968*). Our study indicated that YLD showed positive correlations with HT in dry environments, suggesting that taller plants performed better under dry environments. When accompanied by high temperatures, a severe drought lowers yields in nearly all crops in water-limited production agriculture (*Hossain et al., 2012*). However, plants utilize different drought tolerance mechanisms to sustain yield under drought stress. In a dry environment, early maturing cultivars were able to avoid drought and terminal heat stress and maintained a higher yield, as seen by negative associations between yield and heading date in most drier environments from this study. Also, late-flowering genotypes were disadvantageous under drought since there is less chance of setting florets but a higher chance of being sterile spikelet.

In this study, 115 unique QTL were identified on all the chromosomes except 4A and 5D (Table S4). Among them, 51 consistent QTL and 10 pleiotropic QTL were identified. To validate the QTL found in this study, we compared with some phenological development genes and QTL recently published for these traits. Data from a subset of 11 environments in this study were used to map QTL for yield component traits like kernel per spike (KPS), spike m$^{-2}$ (SPM), and thousand kernel weight (TKW) (*Yang et al., 2020b*). QTL for end-use quality traits were mapped based on a subset of seven environments (*Dhakal et al., 2021*). Comparing QTL mapped with the previously published research based on the same population, eight consistent QTL were confirmed for yield including the ones on 1A at 411.7 Mb, 1D at 421.8, 4B at 659.2 Mb, on 4D at 20.6 and 109.8 Mb, on 6A at 12.4 Mb, on 7B at 15.6 Mb, and on 7D at 591.2 Mb. Furthermore, a set of seven pleiotropic QTL were identified including the ones on 1A at 485.2 Mb for test weight, midline peak width, and midline right slope; two QTL on 2D at 16 Mb for HD and TKW, and at 486.8 Mb for test weight and TKW; three QTL on 4D at 20.6 Mb for biomass yield, yield, and height, at 29.0 Mb for flour protein, harvest index, and test weight, at 109.8 Mb for grain yield, flour yield, flour protein, and test weight; and one QTL on 6D at 459.2 Mb for biomass yield and test weight (*Dhakal et al., 2021*; *Yang et al., 2020b*).

The QTL on 7D at 64.3 Mb for yield and height is close to a QTL for flour yield linked to Excalibur_c22419_460 on 7D at 67 Mb from RL4452 (*Cabral et al., 2018*). Compared with those major genes based on the linked SNPs (*Rasheed et al., 2016*), we found that the QTL on 1D at 421.8 Mb for test weight and yield was about 10 Mb away from the high molecular weight *Glu-D1b* that was located around 412.1 Mb (*Dhakal et al., 2021*). The QTL for yield on 7D at 591.2 Mb was very close to the greenbug resistance gene *Gb3* and *Gb7* (*Liu et al., 2014*; *Tan et al., 2017*). Several QTL for test weight identified from this study were very close or overlapped with other published QTL for quality traits. The test weight QTL on 1D at 375.4 Mb was very close to a QTL linked to Kukri_c20062_389 on 1D at 379.5 Mb (*Jin et al., 2016*). The test weight QTL on 1A at 11.8 Mb was very close to a QTL for midline peak time of dough mixograph linked to RFL_Contig1118_65 at 14.5 Mb

and another QTL on 6A at 603.3 Mb was not far away from a QTL for flour protein linked to Excalibur_rep_c69981_75 at 595.6 Mb of two Chinese wheat cultivars (*Guo et al., 2020*). Four QTL from this study were very close to QTL for quality traits from an association analysis based on nine quality traits (*Yang et al., 2020a*). QTL on 1A at 585.6 Mb for yield was very close to a QTL at 584.7 Mb for grain protein, total starch content, and dough development time; QTL for yield on 4B at 659.2–660.9 Mb was very close to two QTL at 651.8 and 660.7 Mb for grain protein, flour yield, test weight, and wet gluten; QTL for test weight and yield on 6A at 603.3 and 608.5 Mb were very close to a QTL at 602.9 Mb for grain protein, test weight, and total starch content; Compared with QTL found using more than 3,000 lines with more than 50 trait-environment combinations (*Juliana et al., 2019*), three QTL were found at very close physical locations. Yield QTL *Qyld.tamu.1A.586* was very close to a QTL at 585.7 Mb for test weight; *Qtw.tamu.6B.559* was very close to a QTL at 552.9 Mb for thousand kernel weight; QTL associated with test weight, heading date, and yield on 7B at 8.5 and 15.6 Mb were close to a QTL at 8.4 Mb that was around *Vrn-B3* gene; QTL for height, test weight and yield on 7D at 60.6 and 64.3 Mb from this study were close to a QTL for maturity time where *Vrn-D3* was around. We identified *Qyld.tamu.1A.587* for YLD with peak marker IWB34513, which is very close to peak marker IWA1644 linked to early flowering gene *Elf3* at 590 Mb (*Alvarez et al., 2016*; *Zhang et al., 2018*). Early flowering is a drought escape mechanism adopted by many crop plants to avoid water-deficit stress. In the US Southern High Plains where moderate-to-severe water stress frequently occurs, early flowering would be a helpful strategy to cope with water-deficit stress. Early maturity achieved through early flowering and maturity resulted in positive genetic gains (*De Vita et al., 2007*).

Pleiotropic QTL detected from various analyses indicated their reliability. Our results showed that 51 consistent QTL were distributed on 15 chromosomes and 10 of them had pleiotropic effects. QTL for strongly correlated traits were often clustered in the same genomic region. However, in this study, we found QTL for traits with weaker correlations also clustered in some cases. For example, QTL for HD and TW on 1A at 12 Mb, and TW and YLD on the long arm of chromosome 1D at 422 Mb. Traits with weak or no correlation can be selected with the combinations of various alleles for improving multiple traits. It also avoids the undesirable effect of one QTL by selecting against it while improving another QTL.

The total of 28 environments for yield were categorized into four mega-environments and 19 environments for TW, 11 environments for height and 10 environments for heading date were classified into two mega-environments, respectively based on the biplot analyses (Fig. 1). TAM 112 alleles increased HD in most QTL while only half of QTL increased height with TAM 112 alleles. TW from 16 out of 32 QTL were increased by TAM 112 alleles that were located on chromosomes 1D, 2B, 4D, 5A, and 6B while yield from eight out of 18 QTL was increased by TAM 112 alleles that were on chromosomes 1B, 4B, and 7D (Table 3, Table S4). TAM 112 alleles increased both yield and test weight under drier environments while TAM 111 alleles increased both traits under irrigated environments (Table 3, Table S4). Similar findings were observed in a greenhouse study using TAM 111 and TAM 112. TAM 112 was able to yield more grains than TAM 111 in dry environments

due to its superior gas exchange efficiency and other genetic differences in the two cultivars (*Chu et al., 2021*; *Reddy et al., 2014*; *Thapa et al., 2018*).

The total PVE was partitioned into PVE due to Additive and additive-by-environment effects, as well as epistasis and epistasis-by-environment interactions; therefore, only those major QTL with larger PVE from additive effects are of interest for further applications while those QTL with larger additive-by-environment and other interactions can be avoided in the future research.

# CONCLUSIONS

A set of 124 recombinant inbred lines derived from a cross of two popular hard red winter wheat cultivars, TAM 111 and TAM 112, was characterized for yield, test weight, height, and heading date from 28 environments during five growing seasons. All the traits had high heritability with most of the phenotypic variations due to genotypic effects. A total of 115 unique QTL were detected for all the traits with 51 consistent QTL were defined. A set of 10 QTL consistent on chromosomes 1A, 1D, 4B, 4D, 6A, 7B, and 7D had pleiotropic effects. Seven QTL for yield and six QTL for test weight that explained phenotypic variations more than 5% with major additive effects was worthy of further applications. Allele from TAM 112 were expressed in the dry environments and TAM 111 in the irrigated environments. Only a few major QTL, three for yield, six for test weight, and one for height, were involved in interactional effects. Because of the complex inheritance of these traits, it will be necessary to validate these QTL in different wheat backgrounds evaluated under similar growth conditions before developing markers for marker-assisted selection.

# ACKNOWLEDGEMENTS

The authors appreciate the technical support from Maria Pilar Fuentealba, Hangjin Yu, and Lisa Garza, and Jaqueline Avila. We thank Shuhao Yu for his review suggestions.

### Funding

This research was supported by Bayer AG and the BASF Corporation, Texas Wheat Producer Board, Texas A&M AgriLife Research, Tom B. Slick Fellowship from Texas A&M University, USDA National Institute of Food and Agriculture (2019-67013-29172 and 2017-67007-25939 to Shuyu Liu). The open access publishing fees for this article have been covered by the Texas A&M University Open Access to Knowledge Fund (OAKFund), supported by the University Libraries. The funders had no role in study design, data collection and analysis, decision to publish, or preparation of the manuscript.

### Grant Disclosures

The following grant information was disclosed by the authors:
Bayer AG and the BASF Corporation.
Texas Wheat Producer Board.

Texas A&M AgriLife Research.
Texas A&M University.
USDA National Institute of Food and Agriculture: 2019-67013-29172, 2017-67007-25939.
The Texas A&M University Open Access to Knowledge Fund (OAKFund).
The University Libraries.

## Competing Interests

Charles D. Johnson is an Academic Editor for PeerJ.

The authors declare there are no competing interests.

## Author Contributions

- Smit Dhakal and Xiaoxiao Liu performed the experiments, analyzed the data, prepared figures and/or tables, authored or reviewed drafts of the paper, and approved the final draft.
- Chenggen Chu analyzed the data, prepared figures and/or tables, authored or reviewed drafts of the paper, and approved the final draft.
- Yan Yang analyzed the data, prepared figures and/or tables, and approved the final draft.
- Jackie C. Rudd and Amir M.H. Ibrahim conceived and designed the experiments, performed the experiments, analyzed the data, authored or reviewed drafts of the paper, and approved the final draft.
- Qingwu Xue performed the experiments, analyzed the data, authored or reviewed drafts of the paper, and approved the final draft.
- Ravindra N. Devkota Jason A. Baker, Shannon A. Baker, Bryan E. Simoneaux, Geraldine B. Opena, Russell Sutton, Kirk E. Jessup and Kele Hui performed the experiments, prepared figures and/or tables, and approved the final draft.
- Shichen Wang, Charles D. Johnson and Richard P. Metz performed the experiments, analyzed the data, prepared figures and/or tables, and approved the final draft.
- Shuyu Liu conceived and designed the experiments, performed the experiments, analyzed the data, prepared figures and/or tables, authored or reviewed drafts of the paper, and approved the final draft.

## Data Availability

The Supplemental Files include all the marker sequences as well as the top 20 line yield across individual environment that will be published.

## Supplemental Information

Supplemental information for this article can be found online at http://dx.doi.org/10.7717/peerj.12350#supplemental-information.

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
