# Peer review of "Genome-wide QTL mapping of yield and agronomic traits in two widely adapted winter wheat cultivars from multiple mega-environments"

_PeerJ, doi:10.7717/peerj.12350_

## Round 0.1 · original submission · Major Revisions

Please revise your manuscript to address the reviewers' concerns, and submit a new version at your early convenience. Thanks.

Reviewer 1 ·

Basic reporting

1. The report is too long and hence difficult to grasp the main points. Please try to shorten the volume. In the result section for the MTAs, just focus on yield. And you may add those MTAS which are common with yield, days to heading, plant height and TKW. At the end of the day, we will not use any of the QTLs which are not linked to yield for practical breeding purposes.
2. Too many analysis and supplementary data have been presented. But in fact, the main Tables and graphs ( Tables S1, S2) and Figure S2 should be part of the main document.
3. A table showing the yield performance of the top 20 lines along with the parents across the 28 environments could be presented.

Experimental design

No comment

Validity of the findings

1. too many analysis but lacks clear reporting. Some of the Tables and Figures should be part of the main document instead of Supplementary.
2. The conclusion part is weak. There are many reports that could lead to a clear recommendation along with the current result. Tell us specifically the MTAs which could be used in marker assisted breeding for yield under moisture stress.

Additional comments

Please try to shorten the volume. In the result section for the MTAs, just focus on yield. And you may add those MTAS which are common with yield, days to heading, plant height and TKW. At the end of the day, we will not use any of the QTLs which are not linked to yield for practical breeding purposes.
2. Too many analysis and supplementary data have been presented. But in fact, the main Tables and graphs ( Tables S1, S2) and Figure S2 should be part of the main document.
3. A table showing the yield performance of the top 20 lines along with the parents across the 28 environments could be presented.

·

Basic reporting

No comment.

Experimental design

No comment.

Validity of the findings

No comment.

Additional comments

This study by Dhakal et al. identified QTLs associated with yield, test weight, heading date and plant height across 28 diverse environments, followed by analyses of QTL pleiotropy and interactions of epistasis, epistasis-by-environment, and additive-by-environment. The field experiments were well designed and phenotypic data were collected and processed appropriately. The consistent QTLs detected in multiple environments would be useful for future cloning and breeding use. This paper was written in a straightforward way and well structured. Care must be taken to correct some minor language problems. In addition, I think that Tables S2 and S3 showed important information so may be included in the main text. Fig. S1c,d are not clear regarding the descriptive statistics.

---

## Round 0.2 · accepted · Accept

I have reviewed the revised manuscript and their responses to reviewers’ comments. I think it is not necessary to return it to the reviewers.

Thanks for your revisions, and hope that you continue to submit your work to PeerJ.